# Does Supply Chain Concentration Affect the Performance of Corporate Environmental Responsibility? The Moderating Effect of Technology Uncertainty

**Tingli Liu [1] and Hongqiao Gao [2],***

[1] College of Economics and Business Administration, Beijing University of Technology, Beijing 100124, China; liutingli@bjut.edu.cn

[2] Economics and Management College, Civil Aviation University of China, Tianjin 300300, China

* Correspondence: gaohq0903@163.com

**Abstract:** With the development of society and the improvement of environmental consciousness, the performance of corporate environmental responsibility (CER) has elicited increasing attention in recent years. In previous studies, the exploration of the antecedents of CER is far less evident than the exploration of its results, and only few studies have investigated what determines CER engagement from the perspective of supply chain concentration (SCC). Using data from 2413 firms in China from 2013 to 2019, our study uses the fixed effect model and performs multiple robustness tests to examine the impact of SCC on the fulfillment of CER, its transmission mechanism, and the moderating role of technology uncertainty (TU). Empirical results show that SCC has a pivotal negative impact on CER performance, wherein both supplier concentration (SUP) and customer concentration (CUS) are detrimental to CER performance. Further mechanism analysis shows that such negative effect can be explained by the adverse effect of SCC on the operating cash flow (OCF), in which OCF has a partial mediating effect. Moreover, the negative impact of SCC on CER performance is more significant when the uncertainty of firms' technological environment is stronger. Our study opens the transmission "black box" between SCC and CER performance and incorporates the behaviors of firms, inter-firm relationships, and environmental factors into the same research framework, and provides a theoretical guidance for management practices.

**Keywords:** supply chain concentration; corporate environmental responsibility; operating cash flow; technology uncertainty

## 1. Introduction

With the recent intensification of environmental problems such as the depletion of natural resources, air pollution, water pollution and shortage, and soil erosion, the balance between environmental protection and economic development has attracted worldwide attention, and both green economy and sustainable development have gradually become the future economic development directions of all countries in the world. The COVID-19 virus spread throughout the world since its discovery in late 2019. Many experts show that poor ecological environment is an important factor that affects the generation and spread of virus and the increasing mortality rates [1], hence stressing the importance and urgency of environmental protection. As micro-entities of national economic operation, enterprises are also subjects of natural resource consumption and ecological pollution and have an undeniable responsibility toward environmental protection. Improving the ecological environment should ultimately be integrated into the corporate environmental responsibility (CER) of firms.

Compared with a large number of corporate social responsibility (CSR) studies in the fields of management and economics, CER has been relatively ignored in the literature [2]. Meanwhile, the existing works on the driving factors of CER are less numerous than the

impact studies [3,4] and have focused on external pressures, such as legal and institutional factors [5,6]. Government attention and relevant laws and regulations do play an important role in promoting CER performance, but direct government control does not necessarily lead to better results than market solutions [7]. As a node in the supply chain, companies can consider improving CER performance through good supply chain management [8,9], but few studies have examined the driving factors of CER from the supply chain perspective.

Many companies around the world have started to underscore the relationship orientation of economic development, which has a distinctive reliance on major suppliers or customers. For example, according to the China Stock Market and Accounting Research (CSMAR) database, almost all of the top five suppliers (customers) of Chinese listed companies from 2013 to 2019 had 30% or more annual purchase ratios (sales ratios). In addition, since the 1990s, American companies have gradually changed their previous practice of relying on quantities of customers and suppliers and sought to deal with fewer customers and suppliers. Relevant statistics show that more than one-third of the annual sales revenue of US manufacturing companies come from their few top customers. Therefore, supply chain concentration (SCC) is an issue that cannot be ignored in the field of supply chain management. Previous studies pointed out that SCC has a significant impact on business operations and financial conditions [10,11], so it is also very likely to be an important factor that affects CER, but related research remains scarce. From a traditional operations management (OM) perspective, maintaining close relationships with suppliers (customers) can promote the sharing of information between both parties, thereby improving cooperation efficiency and reducing transaction costs [12,13]. However, SCC may also increase business risks and reduce the negotiation ability of enterprises, which may force them to give up their interests [14,15]. What is the impact of SCC on CER performance? It will be explored in our study.

Our study focuses on the relationship between SCC and CER performance and explores the related impact mechanism. Given that the most basic prerequisite for the fulfillment of CER is to have a material foundation, this paper explains the impact of SCC on CER via the operating cash flow (OCF), which is representative of the "hematopoietic" ability of companies. In addition, the Chinese government attaches great importance to scientific and technological innovation and puts forward that "innovation is the first driving force for development." Innovation is also becoming a common practice in society. According to the Statistical Bulletin of National Science and Technology Expenditures in 2020, published by the National Bureau of Statistics, the intensity of R&D investment in China continued to increase in 2020, and corporate R&D accounted for 76.6% of all R&D expenditures, representing a 10.4% increase compared to the previous year. Enterprises have become the mainstays of technological innovation in China, and their innovation momentum continues. Such continuous high-intensity investment has led to a faster technology update rate and an unpredictable trajectory of technological changes. Therefore, all enterprises face a great uncertainty in the technological environment. Given that a company is not only part of the supply chain but also operates in a turbulent environment, the uncertainty of the external technological environment may affect the relationship between SCC and CER performance. Therefore, this paper investigates the moderating effect of technology uncertainty (TU) based on the above research questions. Finally, the existing research on CER has focused on developed countries, so the findings are not necessarily applicable to developing countries [16]. Since China is the largest developing country in the world and its environmental problems have attracted much attention in recent years, this paper selects Chinese companies as the sample.

In sum, this study explores the relationship between SCC and CER and its transmission mechanism by applying fixed effect models and conducting various robustness tests using a sample of Chinese A-share listed firms from 2013 to 2019. In this way, the research framework of "Supply Chain Concentration—Operating Cash Flow—Corporate Environmental Responsibility" is established, and the moderating effect of TU is explored based on real practice. Our research offers several contributions to the supply chain and

environmental responsibility literature. First, previous studies on the driving factors of CER have mostly focused on institutional factors, external pressures, company-specific factors, and managerial characteristics [2,5,17], yet have ignored the impact of SCC on CER from the supply chain perspective. Meanwhile, previous supply chain management studies on the consequences of SCC have focused mainly on financial and operating variables and have rarely considered sustainability issues. By focusing on environmental responsibility issues, this paper points out the negative impact of SCC on CER performance, thereby filling the research gap on both sides. Second, this research explains the negative correlation between SCC and CER performance from the perspective of financial constraints. Previous mechanism explorations that consider financial constraints have often explored external financing constraints instead of the internal operating cash flow (OCF) from the "hematopoietic" capacity perspective. On the basis of specific circumstances, this study explores the mediating effect of OCF. Third, by combining actual conditions, this study incorporates environmental factors, explores the strengthening effect of TU on the negative relationship between SCC and CER performance, and establishes a research framework for studying the relationship between enterprise behavior, inter-firm relationships, and environmental factors, which can yield important insights for enterprise managers.

The rest of this article is organized as follows. Section 2 reviews the related literature. Section 3 develops the research hypotheses. Section 4 presents the data, sample, measures, and empirical model. Section 5 performs panel regressions and provides the analysis results. Section 6 conducts several robustness checks. Section 7 presents the main results, managerial implications, limitations, and potential future research opportunities.

## 2. Literature Review

### 2.1. Corporate Environmental Responsibility

With the globalization and internationalization of environmental issues in recent years, the world has begun to acknowledge the importance of protecting and improving the environment. All sectors of society have paid full attention to environmental protection, and related research on CER, particularly on its impact, has received increasing attention. Having explored the impact of CER on financial performance, corporate value, cost of capital, investment efficiency, corporate risks, etc., Li et al. (2017) [3] found that CER performance positively affects financial performance and is negatively moderated by organizational slack. Li et al. (2020) [18] found that when companies begin to adopt environmental regulations, CER has a negative impact on corporate value. However, after reaching a certain level, CER begins to enhance the corporate value. El Ghoul et al. (2018) [19] investigated manufacturing companies from 30 countries and found that a high CER corresponds to a low cost of equity capital. Lee and Kim (2020) [4] investigated Korean companies and found that these companies can reduce their excessive investments through CER activities and that the degree of market competition can exacerbate such negative relationship. Cai et al. (2016) [20] found that the CER performance of companies can lead to corporate risks, and this reverse relationship is mainly reflected in the manufacturing industry.

However, relatively few studies have explored the antecedents of CER performance. In these studies, the pre-influencing factors mostly focus on (1) formal system factors (e.g., laws and regulations), (2) external pressure (including stakeholder, market, and social pressure), and (3) firm-specific factors and managerial characteristics. Kim et al. (2017) [5] found that CER performance in civil law regions is significantly better than that in common law regions. Dai et al. (2018) [6] found that customer and competitor pressures prompt companies to formulate positive environmental strategies and that such relationship is moderated by the organizational culture. Tsendsure et al. (2021) [17] found that product market competition tends to prevent firms from addressing environmental challenges, but under market competition management capabilities play a positive role in improving corporate environmental practices. Wang et al. (2021) [2] found that gender diversity in the board of directors will increase CER, especially when female board members hold authoritative positions. Orazalin (2020) [21] found that the existence of a sustainability

committee can improve the effectiveness of CSR strategies and enable companies to improve their environmental performance. Although some scholars explored environmental responsibility and green activities from the supply chain perspective, their results had no empirical backing [9,22].

In addition, most existing CER studies are based on western theories, which are rooted in free markets applied to developed countries. Therefore, they may not be fully applicable to developing countries and emerging economies, where the market mechanisms are often inefficient and related legal systems are incomplete [16]. China, as the largest developing country and one of the largest overall carbon emitters in the world, is a good sample for expanding related research. Interestingly, some scholars have noticed the impact of customer concentration [13,23] on CSR performance using Chinese firms as samples, which also inspired the exploration of this article. Considering that CER is not equivalent to CSR, the pre-factors of CER performance warrant further investigation from the supply chain perspective.

*2.2. Outcomes of Supply Chain Concentration*

Supply chain concentration is an important feature of an enterprise's supply chain structure, including two dimensions, supplier concentration (SUP) and customer concentration (CUS), which reflect the degree of dispersion of upstream suppliers and downstream customers, respectively, in the supply chain [24]. We find that previous researches on the impact of SCC have mainly focused on three aspects, namely, economic consequences for firms, corporate management decisions, and corporate capital market performance [10,11,14,15,25,26]. The details are as follows. Economic consequences mainly include SCC and corporate performance, financing capabilities, capital results, and cost management. Kwak and Kim (2020) [10] found a U-shaped relationship between CUS and supplier profitability that weakens along with an increasing equity proportion of insiders involved in company management. Chod et al. (2019) [25] found that retailers with dispersed suppliers obtain less trade credit than those whose suppliers are more concentrated. Meanwhile, corporate management decisions mainly include SCC and accounting decisions, inventory management, corporate innovation, commercial credit supply, and corporate investment. Zhong et al. (2020) [11] revealed a significant inverted U-shaped relationship between CUS and corporate sustainable innovation. Using Chinese firms as examples, Zhang et al. (2020) [15] found that high-SUP firms are inclined to hold more cash due to precautionary concerns. Capital market performance mainly includes SCC and dividend policy, stock price, and stock price collapse risk. Lee et al. (2020) [14] argued that CUS may represent the source of significant cash flow and business risks for supplier firms and found that corporate customer concentration is positively correlated with a stock price crash risk, while government customer concentration is negatively correlated with a stock price crash risk. Cheng et al. (2020) [26] investigated Chinese companies during the COVID-19 crisis and found that a higher degree of SUP was related to more serious stock price declines over the short-term and medium-term windows right after the Wuhan lockdown.

In general, the research on the impact of SCC presents three perspectives. One view is that increasing SCC positively affects companies. Operations management and marketing literature point out that a high concentration of enterprise suppliers (customers) corresponds to a closer relationship between the firm and its suppliers (customers) and is conducive to information sharing among enterprises, which will reduce suppliers' demand uncertainty and promote JIT manufacturing [27]. For customers, coordination and information sharing activities with key suppliers that provide the firm with insights into suppliers' processes, capabilities and constraints, ultimately enable a more effective product and process design, improve the efficiency of goods acceptance and lower the transaction costs. [12] However, the opposite view is that increasing SCC adversely affects enterprises because a higher-SUP (CUS) will drive the overreliance of companies on upstream (downstream) firms. On the one hand, such overreliance will increase the firms' business risk

(e.g., large-scale interruption of raw materials and reduction in sales) and the cost of their equity capital [28]. On the other hand, companies will lose their bargaining power and be forced to make concessions in product prices, trade credit, and so on in supply chain games, which is not conducive to a better financial performance or R&D intensity [29,30]. Some scholars have combined these two perspectives and propose that the impact of SCC on companies changes over time. For example, Irvine, Park, and Yıldızhan (2016) [31] found that the relationship between customer concentration and profitability is negatively correlated at the early stage but becomes gradually positive as the relationship matures. In sum, the impact of increasing SCC on enterprises is multifaceted and complex, and, regardless of whether the pros or cons of increasing SCC will vary across different research objects and contents, researchers and managers should analyze specific problems. However, no previous study has specifically explored the relationship between SCC and CER and the related impact mechanism.

In summary, we have briefly reviewed the relevant literature in the above two sections, one about CER's outcomes and antecedents and the other on the outcomes of SCC. We find that the exploration of CER antecedents is still lacking in the supply chain perspective, that prior studies on firm-level outcomes of SCC focus on financial and operating decisions, and that the conclusions are inconsistent. Therefore, this article contributes to the literature by exploring the impact of SCC on the fulfillment of CER and its influence mechanism. Considering China is a very good typification of emerging markets and has been plagued by negative environmental concerns in recent years [23], we will select Chinese companies as our study sample.

## 3. Hypotheses Development

### 3.1. SCC and CER

An interdependent cooperative relationship and a game of competing interests exist between an enterprise and its upstream customers and downstream suppliers. This article analyzes the relationship between SCC and CER performance from these two perspectives. Among them, the analysis of the cooperation relationship focuses mainly on the "dedicated investment" between a company and its suppliers (customers). Dedicated investment refers to the investment of a company in one of its partners that aims to promote a cooperative relationship or maintain an existing cooperative relationship, such as a physical-, location-, or human-dedicated investment. A higher concentration of suppliers (customers) corresponds to a closer relationship between the firm and its suppliers (customers) and a higher likelihood for the firm to invest in dedicated assets [32], which leads to a highly rigid cost structure [33]. Once the supplier (customer) terminates the cooperation, the dedicated assets invested by the company will depreciate considerably, and the company will face high conversion costs, which will increase its business risk [14]. Therefore, a company with high SCC may retain more funds to prevent risks [34], and doing so will reduce the related inputs of CER performance.

Transaction conditions, including transaction price, quantity, and commercial credit, result from the game between the suppliers and the customers. During this game, both the suppliers and the customers face two choices, namely, the concession of benefits or changing the counterparty. If the supplier changes customers, then additional market development is required [35] to find new customers. Similarly, if the suppliers change, then the customer will incur certain search costs when re-finding a suitable supplier. In addition, if the company has dedicated investments in original customers or suppliers, then changing a trading partner will depreciate or waste these investments. In other words, both benefit concessions and changing counterparties will cause the company to suffer certain losses. In the end, the company will choose the option with a relatively small loss.

When the concentration of suppliers (customers) is strong, firms will cooperate with fewer suppliers (customers), and the loss of large suppliers (customers) will have a pivotal negative impact on the continuous operation, financial performance, and other aspects of the firm. In this case, the market development cost or search cost will be higher than

that of firms with a low concentration of suppliers (customers) if the trading partner is changed. Furthermore, a higher concentration of suppliers (customers) corresponds to more specialized assets that the enterprise may have invested [32]. Therefore, more specific investments will be wasted when changing suppliers (customers). In sum, when the concentration of suppliers (customers) is stronger, firms are more likely to be in a disadvantageous position in the game. In line with bargaining power theory, firms are more likely to be forced to make concessions that are detrimental to their interests in order to avoid incurring high losses from changing suppliers (customers), which is not conducive to a better CER performance. More specifically, first, when suppliers (customers) are in a dominant position in the game, their bargaining power increases. Customers in an advantageous position are more capable of requesting the firm to lower their sales prices, thereby reducing their gross profit margin and return on assets [5,36], and suppliers in a dominant position are more capable of improving the purchase cost of the firm [37]. Therefore, the profits of firms with a high concentration of suppliers (customers) are more likely to be occupied by suppliers (customers). Second, commercial credit financing is a significant source of corporate financing that has an important impact on corporate cash flow and is the focus of both parties in the negotiation. When suppliers (customers) have a negotiating advantage, they may shorten the payment term and reduce the credit limit [15,38], whereas the customer may put forward requirements such as increasing the proportion of credit sales and extending the repayment period [11,39]. The concession of firms to these conditions will have a huge negative impact on their cash flow and increase their risk of incurring bad debt. Third, when customers have a bargaining advantage, they may require firms to maintain a fixed amount of inventory to prevent the risk of supply interruption [40]. When suppliers have a game advantage, they may force firms to purchase unnecessary goods by means of bundling. The above two situations will increase the inventory of firms and drive them to invest excessively in commodity production or material procurement, which would relieve the pressure in inventory management. To sum up, the concessions made by firms with inferior negotiations may have an adverse impact on their financial performance, commercial credit, and inventory management, which will discourage them from fulfilling their environmental responsibility and restrict their investment in CER, negatively affecting their CER performance. SCC refers to the concentration of enterprise supply chain partners, including CUS and SUP. Based on the above analysis, both would be detrimental to CER. In our study, SCC represents the average supplier and customer concentration of firms. Therefore, the following hypothesis is proposed:

**Hypothesis 1 (H1).** *An enhancement in SCC will inhibit the fulfillment of CER.*

### 3.2. The Mediating Effect of OCF

Based on the above analysis, the adverse impact of SCC on the fulfillment of CER can be summarized as shown in Figure 1. The transmission path from the perspective of the game relationship clearly shows that the three aspects of corporate bargaining power, commercial credit supply and financing, and inventory management all affect corporate OCF, thereby affecting CER performance. The analysis of H1 reveals that, first, companies with high SCC have a relatively weak bargaining power and are more likely to be squeezed out by suppliers and customers in terms of procurement costs or sales prices. This effect not only reduces the profit margin of the company but also increases the cash outflow of its purchasing activities and reduces the cash inflow of its sales activities, both of which will adversely affect the cash flow of its operating activities. Second, those companies that are at a disadvantaged position in the game are more likely to accept the cash or prepayment conditions proposed by the supplier and the credit sales or the extension of the payment term proposed by the customer. The decrease in commercial credit financing or the increase in commercial credit supply have a huge negative impact on the OCF of the company. Third, when a company maintains a certain amount of inventory due to customer

requirements, producing more products requires higher variable costs, hence leading to an outflow of OCF. However, if this batch of products becomes part of the inventory, then no cash flow back is achieved through product sales. In this case, although corporate profits will not be affected due to the lack of cost and revenue carryover, the OCF will decrease accordingly. When a firm purchases temporarily unnecessary goods under pressure from suppliers, this firm must either store these goods for future use or resell them at a lower price. Either choice will negatively affect the OCF of the firm.

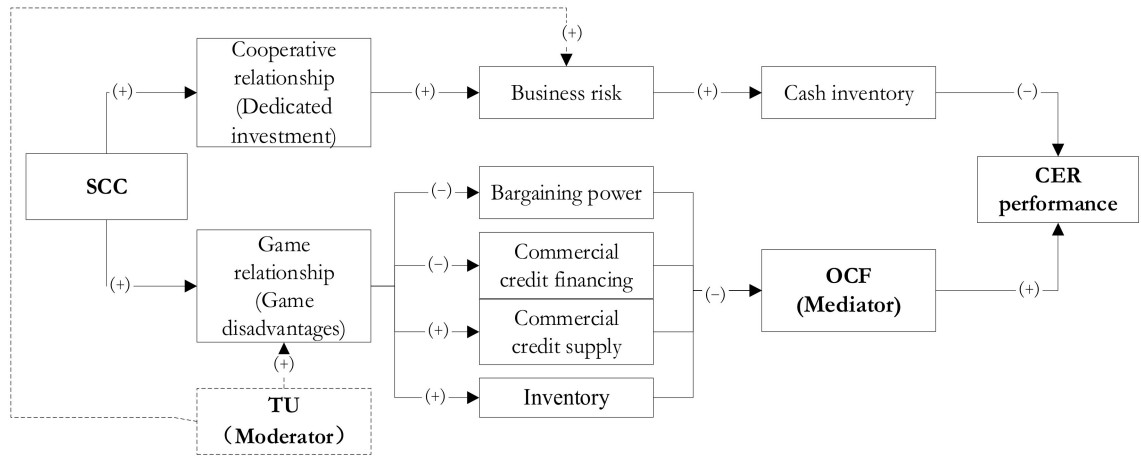

**Figure 1.** Diagram of influence mechanism.

Cash is a manifestation of firm wealth, the blood of firms, and the material basis for firms to fulfill their environmental responsibilities. Reducing the OCF of firms will bring financial constraints on CER performance. Moreover, OCF represents the internal "hematopoietic capacity" of firms. An insufficient "hematopoietic capacity" not only reduces the internal sources of capital of a firm but also sends bad signals to the outside world, which will negatively affect debt and equity financing [41], thereby increasing the financial constraints faced by the company and hinder its fulfillment of CER. Therefore, the following hypothesis is proposed:

**Hypothesis 2 (H2).** *SCC will have an adverse impact on the OCF of firms and consequently reduce their CER performance. OCF has a partial mediating effect on the relationship between the SCC of firms and their CER performance.*

### 3.3. The Moderating Role of TU

Contingency theory posits that organizational decision-making and behavior need to match the external environment. Under the trend of "innovation leads development," technology uncertainty (TU) is a characteristic of the organizational environment that enterprises must face for survival and development and is an essential contingency element to consider for enterprise supply chain management. The TU in this study focuses on the industry in which the company operates. Specifically, TU refers to the unpredictability of the technology development of an industry, including the rapid change, complexity, and difficulty of technology and the continuous creation of new technologies [42]. According to the foregoing analysis, SCC negatively affects CER performance in two ways, namely, (i) by increasing corporate operating risks, which will lead to an increase in cash stock and thereby reduce environmental investment, and (ii) by increasing corporate gaming disadvantages, which increases the concession of benefits and leads to greater losses in OCF, and then increased financial constraints on CER performance. This study suggests that TU may expose corporate suppliers (customers) to greater losses, which will increase the operational risks and enhance the game disadvantages caused by SCC. Therefore, TU moderates the negative relationship between SCC and the fulfillment of CER.

First, as we all know, suppliers and customers willingly cooperate with companies having strong innovation and advanced technologies because these companies not only have higher-quality products but also use their own knowledge and technological achievements to improve the efficiency and effectiveness of the supply chain [43]. For example, in 2021, Dalian Bingshan Group, in China, allowed experts in Japan to complete product acceptance through VR glasses and a remote guidance platform, thereby avoiding the need for customers to travel. When the TU is high, the update and iteration of basic industry knowledge accelerates, and the information and technology that companies have mastered may quickly become out of date. Enterprises are more likely to lose their competitive advantage in the industry, thereby increasing their risk of losing suppliers (customers). On the one hand, such circumstance will increase the risk of devaluation of dedicated investments. Companies with high SCC may be more affected, given that they have more dedicated investments. Therefore, they will retain more cash to prevent risks, which will have a highly significant negative impact on their CER performance. On the other hand, whether because of the loss of competitive advantages compared with peer companies or of a desire to establish close relationships with customers (suppliers) to avoid losing them, firms' negotiating disadvantages due to high SCC are more likely to be strengthened, and suppliers (customers) may require companies to compromise more. This phenomenon will lead to greater losses in OCF, increase the financial constraints faced by a firm, and significantly affect its CER performance. The following hypothesis is then proposed:

**Hypothesis 3 (H3).** *TU can strengthen the negative impact of SCC on CER performance.*

In summary, the research framework of this article is shown in Figure 2.

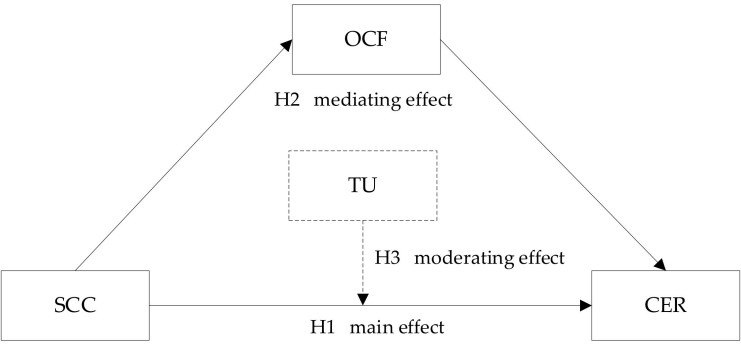

**Figure 2.** Research Framework.

## 4. Methodology

### 4.1. Data and Sample

This study uses Chinese A-share listed firms from 2013 to 2019 as the initial sample. Following prior literature, this paper removes ST shares or *ST shares and excludes firms in the financial industries and B-share (foreign share) firms, given that their regulatory policies and market trading mechanisms are obviously different from those of A-share firms. After excluding firm-years that had missing necessary data for the variables used in the regressions, 2413 companies with 12,351 firm-year observations were left. To eliminate the effect of extreme values, continuous variables with outliers were winsorized based on a 1% quantile tailing. The CER data were collected from the HEXUN database, and all other data were collected from the CSMAR database. HEXUN is one of the largest financial information service providers in China, and its rating systems are credible and have been increasingly adopted by scholars who study CSR's influence on the Chinese context [23,44]. The CSMAR database is one of the largest databases on Chinese listed firms and a major source of credible information on the listed firms' backgrounds and financial statements in China.

*4.2. Variable Measurement*

4.2.1. Measurement of CER Performance

Data on CER performance *(CER)* were collected from HEXUN, a third-party rating agency whose CER evaluation system includes five dimensions, namely, corporate environmental awareness, environmental management system certification, amount of environmental protection investment, number of pollutant types, and number of energy-saving types. A higher score corresponds to a higher level of CER performance.

4.2.2. Measurement of SCC

Following Patatoukas (2012) [27] and Fang et al. (2017) [24], the ratio of the top five suppliers' purchases to the total annual purchases, the ratio of the top 5 customers' sales to the total annual sales, and the averaged sum of these two ratios were used to represent supplier concentration (*SUP*), customer concentration (*CUS*), and supply chain concentration (*SCC*). As sub-indicators, SUP and CUS represent the upstream and downstream concentrations of the supply chain, respectively. Meanwhile, SCC can comprehensively measure the upstream and downstream concentrations in the supply chain of a firm and is helpful in situations where the sub-indexes CUS and SUP are both high, both low, or one high and one low.

4.2.3. Measurement of OCF

Operating cash flow (*OCF*) refers to the cash flow generated by all transactions and events other than the investing and financing activities of a firm. In this study, OCF refers to the net cash flow of a firm from its operating activities, which is computed as the difference between the cash inflow and cash outflow from its operating activities. To eliminate the impact of firm size, this study divided the original net flow by total assets. In addition, to eliminate the influence of magnitude, the value was magnified 100 times for the regression.

4.2.4. Measurement of TU

Chinese patents generally involve a relatively long period of approval, of roughly 18 months. These patents will have an impact on the firm during the application process, and an improvement in the technical strength of a firm does not necessarily involve new inventions and creations. Moreover, not all companies will apply for patents for their own technological achievements. In this case, the patent application and authorization status cannot be used as a good measure of the technological environment. The prerequisite for technological competition among enterprises is a continuously high R&D investment. A higher R&D investment intensity corresponds to a higher likelihood for a firm to have technological output. Therefore, following Ghosh and Olsen (2009) [45], this study adopts the standard deviation of the industry average R&D intensity over the past five years to measure the volatility of the firm's industry technology environment. First, this paper determines the industry classification according to the first code of the 2012 Industry Classification Guidelines for Listed Companies of the China Securities Association and then calculates the total R&D investment and total operating income of all listed companies in each category separately. Second, the ratio of these two variables is calculated to obtain the R&D intensity of each industry in each year. Third, the standard deviation of industry R&D intensity over the last five years is calculated. A higher *TU* value indicates a higher uncertainty in the technological environment of the firm's industry.

4.2.5. Measurement of Control Variables

In order to analyze the impact of SCC on CER performance, following previous studies, this study controlled several other important variables that may affect CER from the perspectives of enterprise size, financial status, corporate governance, and R&D. These control variables include Firm Size (*Size*), Financial Leverage (*Lev*), Operating Income Growth Rate (*Growth*), Return on Assets (*ROA*), Equity Concentration (*Top1*), R&D investment (*R&D*), and Board Size (*Board*). Given that the sample data showed more missing values in

R&D and that having more missing data will largely reduce the sample size, this research referred to the practice of Kim (2018) [46] by setting the missing *R&D investment* values to 0 and by using dummy variables 1 and 0 as substitute variables *(dum_R&D)* to denote whether the R&D variable was missing. The individual ($\delta$) and time ($\omega$) fixed effects were also controlled. The detailed definitions of each variable are shown in Table 1.

**Table 1.** Variables definition.

| Variable Nature | Variables | Name | Definition |
|---|---|---|---|
| Dependent variable | Environmental responsibility performance | CER | Corporate environmental responsibility score report from the HEXUN database |
| Independent variables | Supply chain concentration | SCC | Mean value of CUS and SUP |
| | Customer concentration | CUS | Ratio of sales of the top five customers to total annual sales |
| | Supplier concentration | SUP | Ratio of the purchase amount of the top five suppliers to the total annual purchase amount |
| Mediator | Operating cash flow | OCF | Net cash flow from operating activities/total assets $\times$ 100 |
| Moderator | Technology uncertainty | TU | See above for details. |
| Control variables | Firm size | Size | Natural logarithm of total assets at the end of the period |
| | Financial leverage | Lev | Total liabilities/total assets |
| | Operating income growth rate | Growth | (Increase in operating income this year/total operating income at the end of the previous year) $\times$ 100% |
| | Return on assets | ROA | Net profit/total asset balance |
| | Equity concentration | Top1 | Number of shares held by the largest shareholder/total share capital |
| | R&D investment | R&D | R&D investment/total assets |
| | R&D investment dummy variable | dum_R&D | Equals 1 if R&D investment is missing and equals 0 otherwise |
| | Board size | Board | Expressed by the number of board members |

### 4.3. Model Specifications

To avoid endogeneity, this paper adopts a fixed-effect model that controls for individuals and years and clusters at the firm level to correct the standard error problem caused by serial autocorrelation. In the following model, $\beta_0$ is the intercept, $\delta_i$ is the individual effect, $\omega_t$ is the time effect, and $\varepsilon_{it}$ is the random error. As a vector of control variables, "controls" include *Size, Lev, Growth, ROA, Top1, R&D, Board,* and *dum_R&D*. In addition, to prevent multicollinearity from influencing the conclusions, the variables involving interaction terms were centralized when verifying the moderating effect.

To verify H1, the following model (1) was established, where $x_{it}$ takes the values $SCC_{it}$ to verify the influence of supply chain concentration on the fulfillment of CER. Besides, we took the value of $x_{it}$ to $CUS_{it}$ and $SUP_{it}$ to separately verify the influence of customer concentration and supplier concentration and regarded them as complements of H1. If the influence is negative, then $\beta_1$ in model (1) should be significantly negative.

$$CER_{it} = \beta_0 + \beta_1 x_{it}(SCC_{it}, CUS_{it}, SUP_{it}) + \sum \beta_k controls_{it} + \delta_i + \omega_t + \varepsilon_{it} \qquad (1)$$

To verify the mediating effect of H2, following Baron and Kenny (1986) [47], the stepwise regression method was applied. On the basis of model (1), test models (2) and (3) were constructed. If SCC has a negative effect on OCF, then $\beta_1$ in model (2) should be significantly negative. On this basis, if $\beta_2$ is significant in model (3), then a mediating effect is present. If $\beta_1$ and $\beta_2$ are significant at the same time, then OCF has a partial mediating effect on the relationship between SCC and CER fulfillment. If $\beta_1$ is not significant, then

OCF has a completely mediating effect. If OCF is positively correlated with CER, then $\beta_2$ should be significantly positive.

$$OCF_{it} = \beta_0 + \beta_1 SCC + \sum \beta_k controls_{it} + \delta_i + \omega_t + \varepsilon_{it} \qquad (2)$$

$$CER_{it} = \beta_0 + \beta_1 SCC + \beta_2 OCF + \sum \beta_k controls_{it} + \delta_i + \omega_t + \varepsilon_{it} \qquad (3)$$

To verify H3, on the basis of model (1), the following verification model (4) was constructed. If TU can strengthen the negative effect of SCC on CER fulfillment, then both $\beta_1$ and $\beta_3$ in model (4) should be significantly negative.

$$CER_{it} = \beta_0 + \beta_1 SCC + \beta_2 TU + \beta_3 TU * SCC + \sum \beta_k controls_{it} + \delta_i + \omega_t + \varepsilon_{it} \qquad (4)$$

## 5. Empirical Results

### 5.1. Descriptive Statistics and Correlation

Tables 2 and 3 present the results of the descriptive statistics and correlation analysis of the samples, respectively. As shown in Table 2, the means of CUS and SUP are both greater than 0.3, thereby indicating that for Chinese firms, the purchase of major customers and the sales of major suppliers account for a relatively large portion, and their impacts on the firm warrant further attention. The average CER score of Chinese firms is only 0.977, indicating that the overall CER performance of Chinese enterprises remains very low as of 2019. According to the maximum and minimum data, the CER performance greatly varies from one enterprise to another, thereby highlighting the practical significance of investigating the driving factors of CER fulfillment. The correlation analysis results in Table 3 initially verify the negative correlation between SCC(SUP/CUS) and both OCF and CER performance and the positive correlation between OCF and CER performance. However, this relationship needs to be verified in the regressions. In addition, the results of the three multicollinearity tests with SUP, CUS, and SCC as independent variables reveal that the maximum variance inflation factor (VIF) is less than 5, thereby indicating the absence of any serious multiple cointegration among the explanatory variables.

**Table 2.** Descriptive statistics.

| Variables | Mean | Std. Dev. | Min | Max |
|:---:|:---:|:---:|:---:|:---:|
| CER | 0.977 | 3.794 | 0.000 | 20.000 |
| SCC | 31.699 | 16.014 | 5.000 | 79.950 |
| CUS | 30.158 | 21.284 | 1.160 | 94.430 |
| SUP | 33.311 | 19.252 | 4.780 | 91.570 |
| OCF | 4.667 | 6.723 | −15.672 | 24.163 |
| TU | 0.004 | 0.003 | 0.000 | 0.021 |
| Size | 22.335 | 1.270 | 19.971 | 26.190 |
| Lev | 0.431 | 0.203 | 0.061 | 0.916 |
| Growth | 0.201 | 0.469 | −0.544 | 3.195 |
| Top1 | 33.543 | 14.510 | 8.447 | 72.634 |
| ROA | 0.034 | 0.070 | −0.354 | 0.194 |
| R&D | 0.021 | 0.020 | 0.000 | 0.103 |
| dum_R&D | 0.114 | 0.317 | 0.000 | 1.000 |
| Board | 8.459 | 1.613 | 5.000 | 14.000 |

**Table 3.** Correlation coefficients of the variables.

| | CER | SCC | CUS | SUP | OCF | TU | Size | Lev | Growth | ROA | Top1 | R&D | dum_R&D | Board |
|---|---|---|---|---|---|---|---|---|---|---|---|---|---|---|
| CER | 1 | | | | | | | | | | | | | |
| SCC | −0.071 *** | 1 | | | | | | | | | | | | |
| CUS | −0.051 *** | 0.819 *** | 1 | | | | | | | | | | | |
| SUP | −0.063 *** | 0.774 *** | 0.273 *** | 1 | | | | | | | | | | |
| OCF | 0.040 *** | −0.068 *** | −0.069 *** | −0.040 *** | 1 | | | | | | | | | |
| TU | 0.102 *** | 0.001 | 0.009 | −0.008 | 0.004 | 1 | | | | | | | | |
| Size | 0.144 *** | −0.260 *** | −0.190 *** | −0.228 *** | 0.030 *** | −0.247 *** | 1 | | | | | | | |
| Lev | 0.062 *** | −0.102 *** | −0.054 *** | −0.111 *** | −0.184 *** | −0.172 *** | 0.506 *** | 1 | | | | | | |
| Growth | −0.011 | 0.045 *** | 0.037 *** | 0.035 *** | −0.016 * | −0.020 ** | 0.028 *** | 0.020 ** | 1 | | | | | |
| ROA | 0.036 *** | −0.084 *** | −0.081 *** | −0.054 *** | 0.344 *** | 0.018 * | 0.019 ** | −0.340 *** | 0.177 *** | 1 | | | | |
| Top1 | 0.033 *** | −0.040 *** | −0.023 ** | −0.044 *** | 0.096 *** | −0.056 *** | 0.158 *** | 0.035 *** | −0.012 | 0.146 *** | 1 | | | |
| R&D | −0.103 *** | −0.031 *** | 0.025 *** | −0.083 *** | 0.106 *** | −0.102 *** | −0.132 *** | −0.147 *** | −0.049 *** | 0.031 *** | −0.055 *** | 1 | | |
| dum_R&D | 0.011 | 0.077 *** | −0.002 | 0.139 *** | −0.069 *** | −0.258 *** | 0.084 *** | 0.182 *** | 0.032 *** | −0.057 *** | 0.057 *** | −0.368 *** | 1 | |
| Board | 0.071 *** | −0.097 *** | −0.076 *** | −0.080 *** | 0.036 *** | −0.027 *** | 0.272 *** | 0.143 *** | −0.038 *** | 0.017 * | −0.012 | −0.067 *** | 0.026 *** | 1 |

Note. *** $p < 0.01$, ** $p < 0.05$, * $p < 0.1$.

### 5.2. Impact of SCC on CER

Models (1) to (3) in Table 4 present the regression results regarding the effect of SCC and its two dimensions (CUS and SUP) on CER performance. According to model (1), when the explained variable is CER, the coefficient of SCC is significantly negative ($\beta_1 = -0.013$, $p < 0.01$), thereby indicating that SCC has a negative effect on CER performance. Therefore, H1 is supported. In addition, we explored the effects of two dimensions of SCC—CUS and SUP—on CER and regarded them as complements to H1. As shown in models (2) and (3), the coefficients of CUS and SUP are −0.009 and −0.007, respectively, both of which are significant at the 5% level, thereby indicating that both CUS and SUP are negatively related to CER performance.

### 5.3. Meditating Role of OCF

Models (4) and (5) in Table 4 present the results for the mediation effect of OCF obtained via stepwise regression. According to model (4), when the explained variable is OCF, the coefficient of SCC is significantly negative ($\beta_1 = -0.028$, $p < 0.01$), thereby indicating that, under the influence of both suppliers and customers, SCC has an adverse effect on the OCF of firms. According to model (5), the coefficient of OCF is significantly positive ($\beta_2 = 0.010$, $p < 0.1$), thereby indicating that OCF can support and positively affect CER performance. Combining models (1), (4), and (5) reveals that OCF has a mediating effect on the relationship between SCC and CER performance. Given that the coefficient of SCC in model (5) remains significantly negative ($\beta_1 = -0.013$, $p < 0.05$), OCF exerts a partial mediating effect, thereby supporting H2.

**Table 4.** Regression analysis results.

| Variables | Test of Main Effect | | | Test of Mediating Effect | | Test of Moderating Effect |
| | (1) | (2) | (3) | (4) | (5) | (6) |
| | CER | CER | CER | OCF | CER | CER |
|---|---|---|---|---|---|---|
| SCC | −0.013 *** | | | −0.028 *** | −0.013 ** | −0.012 ** |
| | (0.005) | | | (0.010) | (0.005) | (0.005) |
| CUS | | −0.009 ** | | | | |
| | | (0.004) | | | | |
| SUP | | | −0.007 ** | | | |
| | | | (0.003) | | | |
| OCF | | | | | 0.010 * | |
| | | | | | (0.006) | |
| TU | | | | | | −0.466 |
| | | | | | | (25.397) |
| TU*SCC | | | | | | −2.762 *** |
| | | | | | | (1.049) |
| Controls | Y | Y | Y | Y | Y | Y |
| Year FE | Y | Y | Y | Y | Y | Y |
| Firm FE | Y | Y | Y | Y | Y | Y |
| R-squared | 0.139 | 0.139 | 0.139 | 0.047 | 0.139 | 0.140 |
| N: 12,351; Number of Firm: 2413 | | | | | | |

Notes: *** $p < 0.01$, ** $p < 0.05$, * $p < 0.1$. The cluster-robust standard errors are outlined in parentheses. For brevity, the estimated intercept and control variables are not reported. Before constructing the interaction terms (TU*SCC), we mean-centered the variables.

*5.4. Moderating Role of TU*

Model (6) in Table 4 shows the moderating effect of TU on SCC and CER performance. According to model (6), when the explained variable is CER performance, the coefficient of SCC is significantly negative ($\beta_1 = -0.012$, $p < 0.05$), and the coefficient of the interaction term (TU*SCC) between SCC and TU is significantly negative ($\beta_3 = -2.762$, $p < 0.01$), thereby indicating that the TU increases the negative impact of SCC on CER performance, which supports H3. A visual analysis is presented in Figure 3. A high level of TU corresponds to a greater absolute value of the slope in the image, that is, the negative impact of SCC on CER performance becomes more significant.

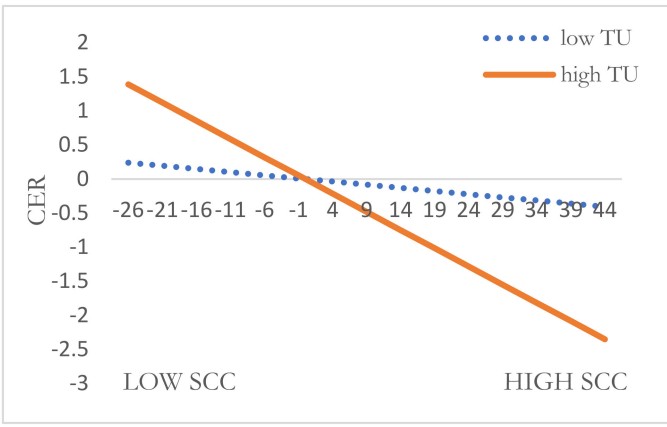

**Figure 3.** Moderating effect of TU on SCC and CER.

## 6. Endogeneity and Robustness Test

### 6.1. Endogeneity Test: Heckman Two-Step Method

The SCC data are collected from corporate reports, but not all firms disclose information about their major suppliers and customers. From the perspective of the interaction logic between SCC and CER performance, the government, investors, creditors, and other stakeholders have paid increasing attention to CSR performance. Therefore, firms that perform less environmental activities may seek to disclose more information about the other aspects of their performance to increase their CSR rating, such as information about their major suppliers and customers. Therefore, the above study may face sample selection bias, which will be addressed using the Heckman two-step method. Given the unavailability of a *Stata* command that allows panel data to control individuals, industry and year dummy variables are added in the regression models, as in models (5) and (6), to control for the industry and time fixed effects. Equation (5) represents the probit SCC selection model for the first stage of Heckman, $dum_{SCC}$ equals 1 if the company discloses its SCC and equals 0 otherwise, and $Pr(dum_{SCC} = 1)$ represents the probability that the company discloses its SCC. Model (6) is the second-stage Heckman model that adds the inverse Mills ratio ($\lambda$) obtained from model (5) to overcome the sample selection bias.

$$Pr(dum_{SCC} = 1) = \varnothing(\gamma Z_{it}) = \beta_0 + \beta_1 SCC_{it-1} + \beta_3 * CER_{it-1} + \sum \beta_k controls_{it} + \mu_{industry} + \mu_{year} + \varepsilon_{it} \qquad (5)$$

$$CER_{it} = \beta_0 + \beta_1 SCC_{it} + \rho\sigma\hat{\lambda}(\gamma Z_{it}) + \sum \beta_k controls_{it} + \mu_{industry} + \mu_{year} + \varepsilon_{it} \qquad (6)$$

The regression results are shown in Table 5. Model (1) is the Heckman one-stage regression result. The coefficient of inverse Mills ratio ($\lambda$) is significant at the 1% level, thereby indicating the presence of a sample selection bias and suggesting that using the Heckman two-step method is appropriate for the analysis. Model (2) presents the second-stage regression, whose results show that SCC has a negative impact on CER performance, which is consistent with the aforementioned conclusions.

### 6.2. Robustness Test

#### 6.2.1. Increase Control Variables

To avoid the endogenous bias caused by the omitted variables, the number of control variables is increased. Given that the development status and background of an enterprise will vary along with its age *(Age)*, "old-brand" enterprises are more likely to have a sense of social responsibility and to perform more environmental activities. Therefore, the control variable corporate age *(Age)* is added. In addition, given the potential impact of internal control and external institutional investors on CER performance [48,49], the number of internal control and external governance variables is also increased by adding the separation of two positions *(Dua)* and the proportion of institutional investors *(Institution)*. *Age* represents the number of years since the establishment of an enterprise. *Dua* is a dummy variable that equals 1 if a chairman of the firm also serves as the CEO and equals 0 otherwise. *Institution* denotes the number of shares held by institutional investors as a share of the total number of shares. The regression results for the fixed year effect and individual effect are shown in models (1) to (3) in Table 6. SCC and supplier/customer concentration inhibit CER performance, which is consistent with the aforementioned conclusions.

**Table 5.** Endogeneity test results.

| Variables | Heckman Two-Step Method | |
|---|---|---|
| | **(1)** | **(2)** |
| | $dum_{SCC}$ | **CER** |
| SCC | | −0.008 ** |
| | | (0.004) |
| $SCC_{it-1}$ | 0.010 *** | |
| | (0.002) | |
| $CER_{it-1}$ | 0.017 ** | |
| | (0.007) | |
| $\lambda$ | −6.328 *** | |
| | (1.315) | |
| Controls | Y | Y |
| Year FE | Y | Y |
| Industry FE | Y | Y |
| Observations | 13,551 | |

Notes: *** $p < 0.01$, ** $p < 0.05$. The robust standard errors are outlined in parentheses. For brevity, the estimated intercept and control variables are not reported.

**Table 6.** Robustness test results.

| Variables | Increase Control Variables | | | Adjust Sample Period | | |
|---|---|---|---|---|---|---|
| | **(1)** | **(2)** | **(3)** | **(4)** | **(5)** | **(6)** |
| | **CER** | **CER** | **CER** | **CER** | **CER** | **CER** |
| SCC | −0.013 ** | | | −0.013 ** | | |
| | (0.005) | | | (0.006) | | |
| CUS | | −0.008 ** | | | −0.008 * | |
| | | (0.004) | | | (0.005) | |
| SUP | | | −0.007 ** | | | −0.007 * |
| | | | (0.003) | | | (0.004) |
| Controls | Y | Y | Y | Y | Y | Y |
| Year FE | Y | Y | Y | Y | Y | Y |
| Firm FE | Y | Y | Y | Y | Y | Y |
| R-squared | 0.139 | 0.139 | 0.139 | 0.143 | 0.143 | 0.143 |
| N | | 12,351 | | | 9889 | |
| Number of Stkcd | | 2413 | | | 2400 | |

Notes: ** $p < 0.05$, * $p < 0.1$. The cluster-robust standard errors are outlined in parentheses. For brevity, the estimated intercept and control variables are not reported. Columns (1) to (3) add the three control variables of *Age*, *Dua*, and *Institution* on the original basis.

### 6.2.2. Adjust Sample Period

In 2015, the General Administration of Quality Supervision, Inspection and Quarantine of China (AQSIQ) and the Standardization Administration of China (SAC) jointly issued a series of national standards on social responsibility for the first time in China. As the country has started to pay more attention to environmental responsibilities, Chinese firms have become more active in fulfilling their environmental responsibilities. To understand whether or not the impact of SCC on CER performance has changed after the issuance of

these standards, this study adjusts the sample period to 2015–2019. The results in models (4) to (6) in Table 6 are consistent with the above conclusions.

### 6.2.3. Mediation Effect Test with Bootstrap Method

Given the limitations of the three-step method in testing the mediating effect proposed by many scholars, this article uses bootstrap self-sampling methods 5000 times for additional testing. Table 7 presents the results after controlling for the control variables described above (including *Age, Dua,* and *Institution*) and the dummy variables of industry and year, which shows that OCF has a partial mediating effect of 6.32%.

**Table 7.** Bootstrap analysis of the mediating effect.

| Bootstrap Test | Effect Size | Boot SE | Boot CI Lower Limit | Boot CI Upper Limit | Relative Effect Size |
|---|---|---|---|---|---|
| Mediating effect | −0.0003 | 0.0001 | −0.0006 | −0.0001 | 6.32% |
| Direct effect | −0.0049 | 0.0022 | −0.0098 | −0.0007 | 93.68% |
| Total effect | −0.0052 | 0.0022 | −0.0101 | −0.0011 | |

## 7. Conclusions and Implications

### 7.1. Main Conclusions

This study investigates the impact of SCC on CER performance and explains such effect from the perspective of capital constraints resulting from the internal "hematopoietic" capacity of enterprises, which opens the "black box" of transmission. This study also examines the moderating effect of TU on the above relationship. To the best of the authors' knowledge, this study is the first to deeply investigate the impact of SCC on CER performance, which so far has been inadequately explored in the literature. Furthermore, building upon the real environment, this study incorporates three variables from micro to macro, namely, enterprise behavior, inter-enterprise relationship, and technological environment uncertainty, into the research framework and expands the supply chain and environmental responsibility literature. Based on a sample of Chinese A-share listed firms from 2013 to 2019, this study reveals several critical findings through fixed effects models and multiple robustness tests. First, SCC has a significant negative impact on CER performance, which is not conducive to the sustainable development of enterprises. Specifically, both SUP and CUS are detrimental to CER performance. Second, SCC has a negative impact on corporate OCF and reduces CER performance by reducing OCF. Moreover, OCF has a partial mediating effect on the relationship between SCC and CER fulfillment, thereby indicating that the concentration of suppliers and customers threatens the "hematopoiesis" ability of a firm and influence its sustainable development decision-making. Third, TU significantly enhances the negative correlation between SCC and CER performance, thereby indicating that in a turbulent technological environment, those firms with high SCC generally face higher risks, attach more importance to their existing partners, and will make more concessions, thereby harming their CER performance.

### 7.2. Managerial Implications

This research offers some valuable insights for the sustainable development of enterprises and their choice of transaction plans. First, managers should pay attention to the SCC of firms. Although strengthening cooperation with major customers or suppliers and establishing close relationships may bring certain benefits to the company, they may also cause the company to lose its bargaining power and introduce constraints and risks to its operations. A concentrated customer or supplier base has a significant negative impact on CER performance, thereby making corporate behavior short-sighted and not conducive to the sustainable development of the company. Therefore, managers should actively expand their channels of suppliers and customers, as well as actively perform the corresponding responsibilities toward existing suppliers and customers, and enhance the

company's reputation and product competitiveness, so as to attract more new partners and maintain an appropriate supply chain concentration. In addition, managers should be careful not to over-compromise on transaction terms such as business credit for fear of losing key suppliers (customers). They should have the courage to break the game dilemma, learn to sacrifice short-term losses for long-term benefits, and avoid the vicious circle of increasing reliance. Second, suppliers or customers with negotiation advantages should use their bargaining power with caution. The competition between enterprises has evolved into a competition between supply chains. The behavior of enterprises is no longer the behavior of independent individuals, but a behavior of mutual influence and interaction in the supply chain network. As the idea of sustainable development attracts increasing attention, the CER implementation of one company in the chain may affect the entire supply chain. For example, the 2015 emissions scandal led to huge penalties for Volkswagen, and the company lost one-third of its market value, which had a significant impact on the market value of its supply chain partners [50]. When suppliers or customers use their own negotiating advantages to reduce corporate cash flow, firms will respond in opportunistic ways, such as by reducing their environmental performance. From a sustainability perspective, such approach has an adverse impact on all companies. Third, when making management decisions, companies should consider the impact of the external technological environment and adjust their trading plans with suppliers or customers in a timely manner according to changes in their technological environment.

### 7.3. Limitations and Future Research Directions

We conclude the study by pointing out some caveats and some directions for future research. First, this research only uses econometric methods to analyze the secondary data of Chinese listed companies. The results of this work may only be applicable to China and have limited applicability in companies from other countries or regions. Multi-method and cross-country research may be conducted in the future to address this limitation. Second, given the limited information on the names and nature of the suppliers and customers of a company, this research only focuses on the proportion of the top five suppliers and customers. However, some interesting phenomena may be left undetected. Future research may check for potential differences in the impact of customer and supplier heterogeneity on CER performance and explore the internal reasons. Third, environmental uncertainty includes market uncertainty, technology uncertainty, economic policy uncertainty, and other aspects, but this study only explores the moderating effect of technology uncertainty. Future investigations may be carried out from the other perspectives of environmental uncertainty.

**Author Contributions:** Conceptualization, T.L. and H.G.; formal analysis, H.G.; investigation, T.L.; data curation, H.G.; writing—original draft preparation, H.G.; writing—review and editing, T.L.; supervision, T.L.; funding acquisition, T.L. All authors have read and agreed to the published version of the manuscript.

**Funding:** This research was funded by the National Social Science Foundation of China, grant number 18BGL090.

**Institutional Review Board Statement:** Not applicable.

**Informed Consent Statement:** Not applicable.

**Data Availability Statement:** Please contact the corresponding author. The data are not publicly available due to conditional requests from the source.

**Acknowledgments:** The authors would like to thank the editors and anonymous reviewers for their constructive comments and suggestions.

**Conflicts of Interest:** The authors declare no conflict of interest.

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
