# Peer review of "Does Supply Chain Concentration Affect the Performance of Corporate Environmental Responsibility? The Moderating Effect of Technology Uncertainty"

_sustainability, doi:10.3390/su14020781_

Round 1

Reviewer 1 Report

The paper is quite interesting and, in my opinion, brings a sort of a new perspective on environmental issues in the sphere of business. 

I do not notice any fundamental editing errors, the article is clearly composed for me, the hypotheses are understandable, and they do not raise any major objections. I consider the research part to be sufficient and the final conclusions to be clear. Literature is a drawback, meaning its selection because I have the impression that it is not very diverse in terms of nationality.

In my opinion, the part in the text "[...] has also been used in many researches published in top journals such as Management Science and the Journal of Financial Economics'" should be deleted as the information has no scientific value and is not essential.

Reviewer 2 Report

[1] At the end of the literature review, please add a summarized comments on literature.

[2] Figure 1 only shows  the "influence mechanismIt" explained in section 3.2 ,  my suggestions is better in the Figure 1 use the variable names to show the relationships you want to test in the later analysis,  and put it at the end of hypotheses.   

Reviewer 3 Report

Review:

Introduction

Coase Theorem; there is no reference and why is this relevant to this study.

In p. 2, the authors claim:” Ac-cording to the China Stock Market and Accounting Research (CSMAR) database, almost all of the top five suppliers (customers) of Chinese listed companies from 2013 to 2019 had 30% or more annual purchase ratios (sales ratios).’ The sentence is clear, but what the authors mean? How it is linked to this study is unclear.

The authors claim that supply chain concentration (SCC) is important; it is; yet, it is not clear how widespread it is, i.e. if only few companies do it or many companies.

I think the authors confuse supply chain concentration with supply chain integration, in p. 2, e.g., ‘a high level of SCC can promote the integration of in-formation between enterprises and both the upstream and downstream, improve cooperation efficiency, and reduce transaction costs’ although they overlap they are not the same; the authors should explain how they treat these 2 concepts separately.

What is hematopoietic in the context of supply chain?

The research aim is valid: ‘examines the impact of the concentration of suppliers or customers on CER, and explores the related impact mechanism’ however, suppliers or customers are 2 different categories, with different supply chain concentration levels.

In short, the introduction needs some revising to clarify the concepts used and describe the links between them more clearly.

Literature review

The section on Corporate Environmental Responsibility is well developed, however, I would expect to see more studies from/about China which is where this study took place; any differences with other parts of the world? Any recent developments?

2.2. Influence of Supply chain concentration

‘Previous research on the impact of SCC has mainly focused on three aspects, namely, economic consequences for firms, corporate management decisions, and corporate capital market performance’  what is this Previous research? No citation here.

I think a weakness of the review is that the Supply chain concentration is not explained before its influence reviewed.

Further, the authors argue

On the one hand, according to transac-tion cost theory and supply chain management theory, a high concentration of enterprise suppliers (customers) is conducive to information sharing and resource complementation among enterprises, which can promote the supply chain integration of enterprises and improve their turnover rate and return on assets, reduce their transaction costs, reduce information asymmetry, and ensure that these firms generate excess profits

This is a strong statement; it needs more explanation.

This is the same for the following statements: this section needs further development.

 Another weakness is that it focuses on customers and not suppliers; it is an important weakness that needs to be resolved

  1. Hypotheses Development

3.1. SCC and CER

Again this confusion between supply chain concentration and  supply chain integration is apparent: e.g., in p. 5

dedicated investment facilitates supply chain integration and improves operational effi-ciency, such investment also leads to a highly rigid cost structure [25] and increases the business risk of the firm.

The Game relationship and Game disadvantages is quite interesting but all the concepts in Figure 1. Diagram of influence mechanism need better explanation.

The authors test a mediation and moderating effect; however, each link in Figure 1 should be a different hypothesis.

My suggestion is to simplify the hypothesis and make them more concrete especially H1.

Methodology

The authors have developed a large dataset, well done in this, integrating the HEXUN database with the CSMAR database

Conclusion and Implications

Findings are significant. Specifically the finding that SCC has a significant negative impact on CER performance, SCC has a negative impact on corporate OCF and reduces CER performance by reducing OCF, and that TU significantly enhances the negative correlation between SCC and CER performance, thereby indicating that in a turbulent technological environment, those firms with high SCC generally face higher risks, attach more importance to their existing partners and will make more con-cessions, then will fulfill less CER. I think the last one is also a very important contribution especially for the current period of uncertainty.

Regarding Managerial Implications I think it is valid to say that lower integration may solve some problems but the authors should also suggest how to manage existing relationships since they have evolved in this way for specific reasons.

Recommendation

The literature review needs to explain the key concepts and hypotheses much better. Some other revisions across the text are required.

......

Title:

 Does supply chain concentration affect the performance of corporate environmental responsibility? The moderating effect of technology uncertainty

Journal: Sustainability

 Abstract: With the development of the society and improvement of environmental consciousness, the performance of corporate environmental responsibility (CER) has elicited increasing attention in recent years. In previous studies, the exploration of the antecedents of CER is far less than the exploration of its results, and only few studies have investigated what determines CER engagement from the perspective of supply chain concentration (SCC). Using data from 2,413 firms in China from 2013 to 2019, our study uses the fixed effect model and performs multiple robustness tests to examine the impact of SCC on the fulfilment of CER, its transmission mechanism, and the moderating role of technology uncertainty (TU). Empirical results show that SCC has a pivotal negative impact on CER performance, wherein both supplier concentration (SUP) and customer concentration (CUS) are detrimental to CER performance. Further mechanism analysis shows that such negative effect can be explained by the adverse effect of SCC on operating cash flow (OCF), in which OCF has a partial mediating effect. Moreover, the negative impact of SCC on CER performance is more significant when the uncertainty of firms’ technological environment is stronger. Our study opens the transmission “black box” between SCC and CER performance, incorporates the behaviors of firms, inter-firm relationships, and environmental factors into the same research framework, and provides a theoretical guidance for management practices.

Keywords: supply chain concentration; corporate environmental responsibility; operating cash flow; technology uncertainty

Round 2

Reviewer 3 Report

I read the revised manuscript and the detailed response to my previous recommendations and I am now convinced that this paper worths publication so I recommend its publication.

....authors response

Response to Reviewer 3 Comments

Point 1: Coase Theorem; there is no reference and why is this relevant to this study. Response 1: Thank you very much for your comment. ‘Direct government control does not necessarily lead to better results than market solutions’ is a point of view of Coase (1960). This statement plays a transitional role in our study. The specific explanation is as follows. Unlike previous studies that focus on institutional and legal aspects to explore the drivers of corporate environmental responsibility (CER), our study innovatively explores the role of supply chain concentration (SCC) on CER by analysing the relationships between firms and their suppliers and customers. In contrast to government regulation, the influence of suppliers and customers on firms’ CER belongs to the effect of market mechanisms. Therefore, this statement is intended to illustrate the importance of the market, thereby transitioning the topic from institutions and laws to the market and then leading to the following point of promoting CER through supply chain management. With inspection, we find some shortcomings in the original presentation, and we have made revisions and added relevant literature.

1.           Reference:

Coase, R. H. The problem of social cost. In Classic papers in natural resource economics. Palgrave Macmillan, London. 1960, 117-118.

Point 2: In p. 2, the authors claim:” According to the China Stock Market and Accounting Research (CSMAR) database, almost all of the top five suppliers (customers) of Chinese listed companies from 2013 to 2019 had 30% or more annual purchase ratios (sales ratios).’ The sentence is clear, but what the authors mean? How it is linked to this study is unclear.

Response 2: Thank you very much for your comment. This sentence and the following statements for American companies are examples to demonstrate that suppliers (customers) dependency is already widespread. On the one hand, such examples can support the phenomenon mentioned in the previous sentence that many companies around the world have supplier (customer) dependencies. On the other hand, such examples use data to illustrate the phenomenon of high SCC presence, thereby proving that SCC is an issue that cannot be ignored in supply chain management. Our previous statement may not have been very clear, and we have revised the article accordingly in this revision. Thank you.

Point 3: The authors claim that supply chain concentration (SCC) is important; it is; yet, it is not clear how widespread it is, i.e., if only few companies do it or many companies.

Response 3: Following to response 2, we use two examples from Chinese and American companies to demonstrate that SCC is an important issue in supply chain management that cannot be ignored. Our previous statement may not have been very clear, and we have revised the article accordingly. Thank you.

Point 4: I think the authors confuse supply chain concentration with supply chain integration, in p. 2, e.g., ‘a high level of SCC can promote the integration of information between enterprises and both the upstream and downstream, improve cooperation efficiency, and reduce transaction costs’ although they overlap they are not the same; the authors should explain how they treat these 2 concepts separately.

Response 4: Thanks for your comment! We have reorganized the analysis and added the related reference. A brief explanation is as follows. Firms with a high SCC are more likely to have a close relationship with its suppliers (customers), which can promote the cooperation and information sharing between both two parties (Zhu et al.,2021), thereby improving cooperation efficiency and reducing transaction costs. For the details of the separate analysis of suppliers and customers, please see the second paragraph in Section 2.2.

2.           Reference:

Zhu, M.; Yeung, A. C.; Zhou, H. Diversify or concentrate: The impact of customer concentration on corporate social responsibility. International Journal of Production Economics. 2021, 240, 108214.

Point 5: What is hematopoietic in the context of supply chain?

Response 5: “Hematopoietic” is a visual representation of operating cash flow in the accounting field. Scholars compare a company to a human body, cash to blood, and operating cash flow (OCF) is compared to having hematopoietic ability because it is an important way to generate cash, which are widely used in accounting field papers (e.g., Rujoub et al.,1995; Qin et al.,2020).

3.           References:

  • Rujoub, A.; Cook, D. M.; Hay, L. E. Using cash flow ratios to predict business failures.

Journal of Managerial Issues. 1995, 75-90.

  • Qin, ; Huang, G.; Shen, H.; Fu, M. COVID-19 pandemic and firm-level cash holding— moderating effect of goodwill and goodwill impairment. Emerging Markets Finance and Trade. 2020, 56, 2243-2258.

Point 6: The research aim is valid: ‘examines the impact of the concentration of suppliers or customers on CER, and explores the related impact mechanism’ however, suppliers or customers are 2 different categories, with different supply chain concentration levels. In short, the introduction needs some revising to clarify the concepts used and describe the links between them more clearly.

Response 6: Following your suggestions in this point and point 13, we have simplified H1 to “An enhancement in SCC will inhibit the fulfillment of CER”. The tests about the impacts of supplier concentration (SUP) and customer concentration (CUS) on CER are presented in the empirical analysis section and regarded as complements of H1. On the basis of the above changes, we have concreted the research aim and revised the expression. Thank you.

Point 7: The section on Corporate Environmental Responsibility is well developed, however, I would expect to see more studies from/about China which is where this study took place; any differences with other parts of the world? Any recent developments?

Response 7: Thanks for your comments. The literature review on CER presented previously in Section 2.1 is a summary of the current status of research within China. Moreover, to illustrate the reason for choosing Chinese firms as the sample, we have summarized the current status of CER research in terms of nationalities and added some relevant studies in the context of China. For details, please see the last paragraph of Section 2.1. In addition, we added a description of the reasons for sample selection in Section 1.

Point 8: Previous research on the impact of SCC has mainly focused on three aspects, namely, economic consequences for firms, corporate management decisions, and corporate capital market performance’ what is this Previous research? No citation here.

Response 8: Thanks for your comments. This statement is a summary of the existing literature, which is described in detail below with listing some of the relevant literature. To avoid ambiguity, we have fine-tuned this statement and cited literature.

Point 9: I think a weakness of the review is that the Supply chain concentration is not explained before its influence reviewed.

Response 9: Following your suggestion, we have supplemented the explanation of SCC before reviewing the literature. SCC includes two dimensions: supplier concentration (SUP) and customer concentration (CUS). The subsequent literature review on the impact of SCC is carried out from both SUP and CUS. For details, please see Section 2.2. Thank you.

Point 10: Further, the authors argue ‘On the one hand, according to transaction cost theory and supply chain management theory, a high concentration of enterprise suppliers (customers) is conducive to information sharing and resource complementation among enterprises, which can promote the supply chain integration of enterprises and improve their turnover rate and return on assets, reduce their transaction costs, reduce information asymmetry, and ensure that these firms generate excess profits’. This is a strong statement; it needs more explanation. This is the same for the following statements: this section needs further development.

Response 10: Following your suggestion, we have revised the statement and have given more explanation from two aspects of CUS and SUP. For more details, please see the second paragraph in Section 2.2. Thank you.

Point 11: Another weakness is that it focuses on customers and not suppliers; it is an important weakness that needs to be resolved.

Response 11: Thank you very much for the comment. We have corrected this mistake and enriched the related literature on suppliers (Chod et al.,2019; Zhang et al.,2020; Cheng et al.,2020). For details, please see the Section2.2.

4.           References:

  • Chod, J.; Lyandres, E.; Yang, S. A. Trade credit and supplier competition. Journal of Financial 2019, 131, 484-505.
  • Zhang, ; Zou, M.; Liu, W.; Zhang, Y. Does a firm’s supplier concentration affect its cash holding? Economic Modelling. 2020, 90, 527-535.
  • Cheng, L. T.; Poon, J.; Tang, S.; Wang, J. Does Supplier Concentration Matter to Investors During the COV1D-19 Crisis: Evidence from China? Available at SSRN. 2020,

Point 12: Again this confusion between supply chain concentration and supply chain integration is apparent: e.g., in p. 5 ‘dedicated investment facilitates supply chain integration and improves operational efficiency, such investment also leads to a highly rigid cost structure

[25] and increases the business risk of the firm.’

Response 12: Thank you very much for the comment. We have removed some ambiguous words and reorganized the analysis about dedicated investment. For details, please see the Section 3.1.

Point 13: The authors test a mediation and moderating effect; however, each link in Figure 1 should be a different hypothesis. My suggestion is to simplify the hypothesis and make them more concrete especially H1.

Response 13: Thanks for your suggestion. The Figure 1 is a diagram of influence mechanism that briefly summarizes the textual explanation of the Section 3, and is intended to show our reasoning logic more clearly. The multiple links shown in Figure 1, including “cooperative relationship (dedicated investment)” and “Game relationship (Game disadvantages)”, are explanations for the main hypothesis rather than independent hypotheses. In addition, in response to the question that H1 contains three links, we have simplified H1 to “An enhancement in SCC will inhibit the fulfillment of CER”. The tests about the impacts of SUP and CUS on CER are now presented in the empirical analysis section and regarded as complements of H1. Accordingly, we also modified Figure 1. And at last, we have added a Figure 2 to show our research framework.

Point 14: Regarding Managerial Implications I think it is valid to say that lower integration may solve some problems but the authors should also suggest how to manage existing relationships since they have evolved in this way for specific reasons.

Response 14: Following your suggestion, we have added suggestions on how to manage existing relationships through two aspects. For details, please see Section 7.2. Thank you.

I read the revised manuscript and the detailed response to my previous recommendations and I am now convinced that this paper worths publication so I recommend its publication.

....authors response

Response to Reviewer 3 Comments
Response to Reviewer 3 Comments

Point 1: Coase Theorem; there is no reference and why is this relevant to this study. Response 1: Thank you very much for your comment. ‘Direct government control does not necessarily lead to better results than market solutions’ is a point of view of Coase (1960). This statement plays a transitional role in our study. The specific explanation is as follows. Unlike previous studies that focus on institutional and legal aspects to explore the drivers of corporate environmental responsibility (CER), our study innovatively explores the role of supply chain concentration (SCC) on CER by analysing the relationships between firms and their suppliers and customers. In contrast to government regulation, the influence of suppliers and customers on firms’ CER belongs to the effect of market mechanisms. Therefore, this statement is intended to illustrate the importance of the market, thereby transitioning the topic from institutions and laws to the market and then leading to the following point of promoting CER through supply chain management. With inspection, we find some shortcomings in the original presentation, and we have made revisions and added relevant literature.

1.           Reference:

Coase, R. H. The problem of social cost. In Classic papers in natural resource economics. Palgrave Macmillan, London. 1960, 117-118.

Point 2: In p. 2, the authors claim:” According to the China Stock Market and Accounting Research (CSMAR) database, almost all of the top five suppliers (customers) of Chinese listed companies from 2013 to 2019 had 30% or more annual purchase ratios (sales ratios).’ The sentence is clear, but what the authors mean? How it is linked to this study is unclear.

Response 2: Thank you very much for your comment. This sentence and the following statements for American companies are examples to demonstrate that suppliers (customers) dependency is already widespread. On the one hand, such examples can support the phenomenon mentioned in the previous sentence that many companies around the world have supplier (customer) dependencies. On the other hand, such examples use data to illustrate the phenomenon of high SCC presence, thereby proving that SCC is an issue that cannot be ignored in supply chain management. Our previous statement may not have been very clear, and we have revised the article accordingly in this revision. Thank you.

Point 3: The authors claim that supply chain concentration (SCC) is important; it is; yet, it is not clear how widespread it is, i.e., if only few companies do it or many companies.

Response 3: Following to response 2, we use two examples from Chinese and American companies to demonstrate that SCC is an important issue in supply chain management that cannot be ignored. Our previous statement may not have been very clear, and we have revised the article accordingly. Thank you.

Point 4: I think the authors confuse supply chain concentration with supply chain integration, in p. 2, e.g., ‘a high level of SCC can promote the integration of information between enterprises and both the upstream and downstream, improve cooperation efficiency, and reduce transaction costs’ although they overlap they are not the same; the authors should explain how they treat these 2 concepts separately.

Response 4: Thanks for your comment! We have reorganized the analysis and added the related reference. A brief explanation is as follows. Firms with a high SCC are more likely to have a close relationship with its suppliers (customers), which can promote the cooperation and information sharing between both two parties (Zhu et al.,2021), thereby improving cooperation efficiency and reducing transaction costs. For the details of the separate analysis of suppliers and customers, please see the second paragraph in Section 2.2.

2.           Reference:

Zhu, M.; Yeung, A. C.; Zhou, H. Diversify or concentrate: The impact of customer concentration on corporate social responsibility. International Journal of Production Economics. 2021, 240, 108214.

Point 5: What is hematopoietic in the context of supply chain?

Response 5: “Hematopoietic” is a visual representation of operating cash flow in the accounting field. Scholars compare a company to a human body, cash to blood, and operating cash flow (OCF) is compared to having hematopoietic ability because it is an important way to generate cash, which are widely used in accounting field papers (e.g., Rujoub et al.,1995; Qin et al.,2020).

3.           References:

  • Rujoub, A.; Cook, D. M.; Hay, L. E. Using cash flow ratios to predict business failures.

Journal of Managerial Issues. 1995, 75-90.

  • Qin, ; Huang, G.; Shen, H.; Fu, M. COVID-19 pandemic and firm-level cash holding— moderating effect of goodwill and goodwill impairment. Emerging Markets Finance and Trade. 2020, 56, 2243-2258.

Point 6: The research aim is valid: ‘examines the impact of the concentration of suppliers or customers on CER, and explores the related impact mechanism’ however, suppliers or customers are 2 different categories, with different supply chain concentration levels. In short, the introduction needs some revising to clarify the concepts used and describe the links between them more clearly.

Response 6: Following your suggestions in this point and point 13, we have simplified H1 to “An enhancement in SCC will inhibit the fulfillment of CER”. The tests about the impacts of supplier concentration (SUP) and customer concentration (CUS) on CER are presented in the empirical analysis section and regarded as complements of H1. On the basis of the above changes, we have concreted the research aim and revised the expression. Thank you.

Point 7: The section on Corporate Environmental Responsibility is well developed, however, I would expect to see more studies from/about China which is where this study took place; any differences with other parts of the world? Any recent developments?

Response 7: Thanks for your comments. The literature review on CER presented previously in Section 2.1 is a summary of the current status of research within China. Moreover, to illustrate the reason for choosing Chinese firms as the sample, we have summarized the current status of CER research in terms of nationalities and added some relevant studies in the context of China. For details, please see the last paragraph of Section 2.1. In addition, we added a description of the reasons for sample selection in Section 1.

Point 8: Previous research on the impact of SCC has mainly focused on three aspects, namely, economic consequences for firms, corporate management decisions, and corporate capital market performance’ what is this Previous research? No citation here.

Response 8: Thanks for your comments. This statement is a summary of the existing literature, which is described in detail below with listing some of the relevant literature. To avoid ambiguity, we have fine-tuned this statement and cited literature.

Point 9: I think a weakness of the review is that the Supply chain concentration is not explained before its influence reviewed.

Response 9: Following your suggestion, we have supplemented the explanation of SCC before reviewing the literature. SCC includes two dimensions: supplier concentration (SUP) and customer concentration (CUS). The subsequent literature review on the impact of SCC is carried out from both SUP and CUS. For details, please see Section 2.2. Thank you.

Point 10: Further, the authors argue ‘On the one hand, according to transaction cost theory and supply chain management theory, a high concentration of enterprise suppliers (customers) is conducive to information sharing and resource complementation among enterprises, which can promote the supply chain integration of enterprises and improve their turnover rate and return on assets, reduce their transaction costs, reduce information asymmetry, and ensure that these firms generate excess profits’. This is a strong statement; it needs more explanation. This is the same for the following statements: this section needs further development.

Response 10: Following your suggestion, we have revised the statement and have given more explanation from two aspects of CUS and SUP. For more details, please see the second paragraph in Section 2.2. Thank you.

Point 11: Another weakness is that it focuses on customers and not suppliers; it is an important weakness that needs to be resolved.

Response 11: Thank you very much for the comment. We have corrected this mistake and enriched the related literature on suppliers (Chod et al.,2019; Zhang et al.,2020; Cheng et al.,2020). For details, please see the Section2.2.

4.           References:

  • Chod, J.; Lyandres, E.; Yang, S. A. Trade credit and supplier competition. Journal of Financial 2019, 131, 484-505.
  • Zhang, ; Zou, M.; Liu, W.; Zhang, Y. Does a firm’s supplier concentration affect its cash holding? Economic Modelling. 2020, 90, 527-535.
  • Cheng, L. T.; Poon, J.; Tang, S.; Wang, J. Does Supplier Concentration Matter to Investors During the COV1D-19 Crisis: Evidence from China? Available at SSRN. 2020,

Point 12: Again this confusion between supply chain concentration and supply chain integration is apparent: e.g., in p. 5 ‘dedicated investment facilitates supply chain integration and improves operational efficiency, such investment also leads to a highly rigid cost structure

[25] and increases the business risk of the firm.’

Response 12: Thank you very much for the comment. We have removed some ambiguous words and reorganized the analysis about dedicated investment. For details, please see the Section 3.1.

Point 13: The authors test a mediation and moderating effect; however, each link in Figure 1 should be a different hypothesis. My suggestion is to simplify the hypothesis and make them more concrete especially H1.

Response 13: Thanks for your suggestion. The Figure 1 is a diagram of influence mechanism that briefly summarizes the textual explanation of the Section 3, and is intended to show our reasoning logic more clearly. The multiple links shown in Figure 1, including “cooperative relationship (dedicated investment)” and “Game relationship (Game disadvantages)”, are explanations for the main hypothesis rather than independent hypotheses. In addition, in response to the question that H1 contains three links, we have simplified H1 to “An enhancement in SCC will inhibit the fulfillment of CER”. The tests about the impacts of SUP and CUS on CER are now presented in the empirical analysis section and regarded as complements of H1. Accordingly, we also modified Figure 1. And at last, we have added a Figure 2 to show our research framework.

Point 14: Regarding Managerial Implications I think it is valid to say that lower integration may solve some problems but the authors should also suggest how to manage existing relationships since they have evolved in this way for specific reasons.

Response 14: Following your suggestion, we have added suggestions on how to manage existing relationships through two aspects. For details, please see Section 7.2. Thank you.